# Modeling and Reliability Analysis of MEMS Gyroscope Rotor Parameters under Vibrational Stress

**DOI:** 10.3390/mi15050648

**Published:** 2024-05-14

**Authors:** Lei Wang, Yuehong Pan, Kai Li, Lilong He, Qingyi Wang, Weidong Wang

**Affiliations:** 1School of Mechano-Electronic Engineering, Xidian University, Xi’an 710068, China; lqpyte@163.com; 2Chongqing CEPREI Industrial Technology Research Institute Co., Ltd., Chongqing 401333, China; pyh80_0@163.com; 3China Electronic Product Reliability and Environmental Testing Research Institute, Guangzhou 511370, China; lfhyezi@163.com; 4Xi’an Chuanglian Ultrasonic Technology Co., Ltd., Xi’an 710065, China; lilonghe1980@163.com; 5School of Basic Medicine, Air Force Medical University, Xi’an 710032, China

**Keywords:** MEMS gyroscope, degradation of rotor parameters, reliability evaluation, copula function, vibration environment

## Abstract

Vibrational environments can cause drift or changes in Micro-Electro-Mechanical System (MEMS) gyroscope rotor parameters, potentially impacting their performance. To improve the effective use of MEMS gyroscopes, this study introduced a method for evaluating the reliability of parameter degradation under vibration. We analyzed the working principle of MEMS gyroscope rotors and investigated how vibration affects their parameters. Focusing on zero bias and scale factor as key performance indicators, we developed an accelerated degradation model using the distributional assumption method. We then collected degradation data for these parameters under various vibration conditions. Using the Copula function, we established a reliability assessment approach to evaluate the degradation of the MEMS gyroscope rotor’s zero bias and scale factor under vibration, enabling the determination of reliability for these parameters. Experimental findings confirmed that increasing stress levels lead to reduced failure times and increased failure rates for MEMS gyroscope rotors, with significant changes observed in the zero bias parameter. Our evaluation method effectively characterizes changes in the reliability of the MEMS gyroscope rotor’s scale factor and zero bias over time, providing valuable information for practical applications of MEMS gyroscopes.

## 1. Introduction

With advancements in microelectronics and machining technologies, Micro-Electro-Mechanical System (MEMS) gyroscopes have found widespread applications in aerospace, military, autonomous driving, and other fields [1]. MEMS gyroscopes utilize micro-machining technology to detect the angular velocity of rotating objects. The rotor, a key component of a MEMS gyroscope, is typically fabricated using micro-machining techniques to ensure excellent sensitivity and precision [1,2]. However, harsh environmental conditions, including impact, vibration, humidity, radiation, and corrosion, pose significant challenges to the operational reliability and accuracy of MEMS gyroscopes, resulting in fatigue, fracture, delamination, plastic deformation, particle contamination, sealing degradation, and material outgassing within the gyroscopes [2,3]. Vibration is one of the most common factors that affect the reliability of MEMS devices. The degradation of rotor parameters in MEMS gyroscopes is often attributed to vibration environments [4,5]. Once performance deteriorates to a critical level, the device parameters become unacceptable, leading to measurement errors. Consequently, evaluating the reliability of performance degradation in MEMS gyroscopes is crucial [2].

Previous studies have addressed reliability concerns in MEMS gyroscopes. Gill et al. investigated how harsh environments can affect bias stability, angle random walk, and scale factor [3]. Kang et al. used step-stress accelerated degradation testing (SSADT) to assess a MEMS flow sensor’s reliability under temperature stress, obtaining degradation data for both the chip and the sensor system [4]. Ge et al. studied vibration load characteristics in various gyroscope applications, including mechanical shocks and high temperatures. They used vibration stress step tests to analyze the impact of vibration on MEMS gyroscope performance degradation [5]. Xu et al. introduced a reliability analysis method for MEMS gyroscopes using Monte Carlo simulation. Leveraging the working principle of MEMS gyroscopes, they constructed a frequency matching failure model and developed an additional failure model utilizing the microcantilever beam in MEMS gyroscopes. Based on this simulation, they assessed the reliability and sensitivity of the gyroscopes. The findings indicate that the reliability of the device has a correlation with the size of the cantilever beams and the adhesion energy of the material [6]. Jiang et al. implemented linear regression analysis on the reliability life of a certain type of submarine launched missile gyroscope rotor. They fitted various distribution functions, selected the absolute value of the maximum correlation coefficient as the distribution function, and utilized the determined distribution function and existing sample data to calculate and analyze the reliability index. By leveraging the linearization function formula of the Weibull distribution function, they calculated the reliability level and studied the degradation trend of the gyroscope more effectively [7]. Taking the MEMS accelerometer as the research object, Qin et al. conducted accelerated performance degradation experiments on the accelerometer under vibration stress. Additionally, they carried out an accelerated life reliability assessment and accelerated performance degradation reliability assessment on the failure data of the MEMS accelerometer under a vibration environment, respectively. Their study verified the feasibility of the accelerated performance degradation method in evaluating the reliability of MEMS devices [8]. Leveraging statistical analysis methods based on Copula functions, He et al. analyzed the accelerated degradation data of MEMS accelerometers, obtaining the reliability of MEMS accelerometers under normal vibration conditions. Their study effectively proved that using multiple parameters to evaluate the reliability of MEMS devices yields greater accuracy [9]. The aforementioned studies have explored the reliability of MEMS devices through experimental or simulation approaches. However, there is room for enhancing the accuracy of the evaluation method and refining the selection of characteristic parameters.

Building upon previous findings, this study introduces a method for assessing the reliability of MEMS gyroscope rotor parameter degradation under vibrational conditions, with the goal of enhancing the efficient utilization of MEMS gyroscopes. The paper investigates various manifestations of performance degradation of MEMS gyroscopes during vibration, develops an accelerated degradation model of MEMS gyroscope rotor parameters under vibration, and utilizes the Copula function to assess the rotor reliability.

## 2. Reliability Evaluation of MEMS Gyroscope Rotor’s Parameter Degradation

### 2.1. MEMS Gyroscope Rotor Working Principle

The MEMS gyroscope rotor operates by utilizing the Coriolis effect. This effect causes a displacement or deformation in the rotor—typically designed as a microfabricated cantilever beam or tuning fork [9,10]—when it experiences motion within a rotating reference frame. Capacitive or optical sensing methods detect this change, allowing for the measurement of the Coriolis force and ultimately facilitating the determination of the object’s rotational speed and direction [11,12,13,14]. The control circuit processes and amplifies this signal for transmission to the user or control system. Figure 1 illustrates the schematic representation of the MEMS gyroscope structure.

The MEMS gyroscope rotor generates vibrations during operation. Based on Newton’s second law, the second-order single-degree-of-freedom vibration force balance equation governing the MEMS gyroscope rotor can be expressed as follows:(1)$$F(t)=B\ddot{x}+\psi \dot{x}+\mathrm{kx}$$
where *F*(*t*) represents the inertial force that results from the rotor’s acceleration; *B*$\ddot{x}$ denotes inertial force where *B* is the rotor’s mass; *ψ* is the damping coefficient, with *ψ*$\dot{x}$ indicating the damping force; *kx* is the elastic force; *x*, $\dot{x}$, and $\ddot{x}$ represent the acceleration, its first derivative, and its second derivative, respectively; *k* indicates the nonlinearity of the MEMS gyroscope rotor.

Let us introduce *δ* = $\frac{\psi }{2B}$ as the damping coefficient and $\vartheta_{s}^{2}=\frac{k}{B}$ as the rotor’s natural frequency. Inserting these into Equation (1) gives:(2)$$\mathrm{kF}(t)=\ddot{x}+2\delta \dot{x}+\vartheta_{s}^{2}x$$

The transfer function of the rotor *H*(*s*) can be derived using the Laplace transform of Equation (2):(3)$$H(s)=k\cdot \frac{1}{s^{2}+2\delta s+\vartheta_{s}^{2}}$$

Equation (3) defines *H*(*s*) as the transfer function. By applying a known acceleration to the MEMS gyroscope rotor and performing an inverse Laplace transform, we can derive the rotor’s time-domain response and its deflection behavior. Building upon established relationships between these functions [14,15], we can then determine the sensitivity function of the MEMS gyroscope rotor’s output. This sensitivity function is critical in calculating the rotor’s zero bias and scale factor, which are essential performance metrics.

### 2.2. The Impact of Vibration Environment on MEMS Gyroscopes Rotor’s Parameters 

MEMS gyroscopes find widespread applications in aircraft attitude control and navigation. However, their parameter performance stability is often compromised by vibrations, leading to a decline in gyroscopic accuracy and hindering their ability to accurately depict angular rate signals [12,16,17,18].

Vibrational environments can remarkably influence MEMS gyroscope rotor parameters in several ways. Firstly, vibrations can change the capacitance between the rotor and the gyroscope’s stator, which affects measurement accuracy. Secondly, vibrations can cause mechanical strain within the gyroscope. This strain can change the position of the rotor relative to the stator, affecting measurement sensitivity [19]. 

In environments with strong vibrations, the force on the rotor’s accelerometer structure may exceed its limits, leading to structural damage, such as cracks or breaks. Damage can cause the mass block to touch the gyroscope’s base, which leads to wear [20,21]. Additionally, vibrations can lead to cracks in the encapsulation casing of MEMS accelerometers and detachment of metal leads. The alternating stress from vibrations leads to cumulative damage, causing fatigue failure in MEMS accelerometers [9]. Over time, these changes significantly alter the MEMS gyroscope rotor’s parameters and can make it unusable.

### 2.3. Construction of Accelerated Degradation Model Based on Distribution Assumption

To understand how vibrational conditions impact the deterioration of MEMS gyroscope rotor characteristics, we developed an accelerated degradation model. This model utilized distributional assumptions to predict how rotor parameters change over time under stress [22,23].

To develop an accelerated degradation model for MEMS gyroscope rotors in a vibrational environment, several steps are necessary. First, accelerated degradation testing must be conducted to determine the operational and failure limits for the rotors. The rotor’s zero bias and scale factor will serve as the main indicators of degradation. For this study, failure is defined as occurring when the zero bias value falls outside the range of 2.4 V to 2.6 V or the scale factor falls below 0.3 mV or exceeds 0.5 mV. 

To perform the accelerated degradation tests, 36 rotors were selected and divided into three groups. These groups were subjected to stress levels of 30 g_n_, 40 g_n_, and 50 g_n_. A specialized multifunctional calibration experimental apparatus was used to create the vibrational environment, and the rotor’s output values and scale factor data were carefully recorded during testing.

Based on the output value and scale factor data of the MEMS gyroscope rotor obtained during testing, we formulated acceleration models for zero bias and scale factor. These models used probability distributions to describe how rotor parameters change under stress. We employed the distributional assumption method along with the inverse power law model. In this context, the parameter *η* represents the zero-offset scale of the MEMS gyroscope rotor, and its probability distribution follows the accelerated degradation model expression equation.
ln *η* = ln *Q*(*C*_1_ + *C*_2_)(4)
where *C*_1_ and *C*_2_ are constants from our model describing how the zero bias changes under stress, with *Q* representing the stress level itself.

From Equation (4), we can derive the marginal distribution function of MEMS gyroscope rotor bias:(5)$$F_{1}(t)=1-\exp\left[-{\left(\frac{t}{\eta }\right)}^{m}\right]$$
where *F*_1_(*t*) represents the zero bias distribution function of the MEMS gyroscope rotor; *m* is the distribution assumption parameter; and *t* represents time. Similar to the zero bias model, the scaling factor’s degradation is modeled using probability distributions. The acceleration model under the distributional assumption is described by the following equation:ln *μ* = ln *Q*(*C*_1_′ + *C*_2_′)(6)
where *μ* represents the scale factor; and *C*_1_′ and *C*_2_′ represent the parameters of the scaling factor acceleration degradation model.

Equation (6) allows us to derive the marginal distribution function of the MEMS gyroscope rotor scale factor, expressed as follows:(7)$$F_{2}(t)=\frac{\int_{0}^{t}\exp\left[-\frac{{\left(x-\mu \right)}^{2}}{2\zeta^{2}}\right]dx}{\sqrt{2\pi }}$$

In this equation, the parameter *ζ* represents characteristics of the normal distribution that describes the MEMS gyroscope rotor scale factor. Importantly, Equation (7) provides a way to connect the theoretical model to the degradation data (zero bias and scale factor values) obtained from the accelerated degradation testing in a vibrational environment. Using this degradation data, we can then assess the MEMS gyroscope rotor’s reliability under vibrational conditions.

### 2.4. Reliability Evaluation of MEMS Gyroscope Rotor Parameter Degradation Based on Copula Function

#### 2.4.1. Copula Function for Independent Parameter Failures

We assessed the reliability of MEMS gyroscope rotor parameter degradation using the Copula function, drawing on the degradation data collected during accelerated testing for the rotor’s bias and scale factor. The Copula function is a mathematical tool that connects a multivariate joint distribution function to its corresponding marginal distribution functions [24,25,26]. This allows us to analyze the marginal distributions and the dependence structure between random variables separately, providing a deeper understanding of their relationships and improving model fitting and evaluation. Particularly, the marginal distributions contain complete information about each variable, preventing information loss during the analysis and ensuring more accurate evaluation results. 

Since the failure of either the zero bias or scale factor indicates a failure of the MEMS gyroscope rotor system [27], we defined the failure time for parameter degradation in a vibrational environment as *T* = *min*(*T*_1_, *T*_2_). Here, *T*_1_ represents the failure time of the MEMS gyroscope rotor’s zero bias, and *T*_2_ represents the failure time of the scale factor [28].

When the zero bias and scale factor parameter failures of the MEMS gyroscope rotor occur independently, the Copula function for the rotor’s reliability is expressed as follows:*Z*_1_(*t*) = *P*(*T*_1_ > *ε*, *T*_2_ > *ε*)(8)
where *Z*_1_(*t*) represents the Copula function for the MEMS gyroscope rotor’s reliability under independent failures. The symbol *ε* denotes the failure time threshold for both the zero bias and scale factor parameters, and *P*(·) is the probability function of their failures.

#### 2.4.2. Copula Function for Correlated Parameter Failures

In the context of a vibration environment, the deterioration of parameters related to MEMS gyroscope rotors is frequently interconnected. To analyze this correlation structure, we examined the marginal distributions of the zero bias and scale factor individually. The interrelation among MEMS gyroscope rotor parameters can be elucidated by formulating the joint distribution of random variables [29]. Using Equations (5) and (7), we constructed a Copula function to describe the MEMS gyroscope rotor’s reliability, expressed as:
*H*(*t*) = *G*(*F*_1_(*t*), *F*_2_(*t*); Γ) (9)
where *H*(*t*) is the Copula function for assessing MEMS gyroscope rotor reliability. The function *G*(·) represents the correlation structure that influences the degradation of the rotor’s characteristics, and Γ is a parameter describing the correlation between the rotor’s bias and scale factor [30,31,32].

To address the case of multiple parameter failures in MEMS gyroscope rotors, we extended the Copula function for reliability assessment of parameter degradation, as shown in Equation (10):(10)$$Z_{2}(t)=1-\left[\sum_{i=1}^{n}F_{i}(t)-\sum_{i=1}^{n}H_{i}(t)\right]$$
where *Z*_2_(*t*) represents the Copula function for evaluating the reliability of MEMS gyroscope rotor parameter degradation under multiple failures.

Using Equations (8) and (10), we can obtain reliability assessments for the degradation of MEMS gyroscope rotor parameters in various vibrational scenarios.

## 3. Experimental Analysis

To validate the proposed reliability assessment method for MEMS gyroscope rotor parameter degradation under vibration, we conducted an experimental study using the L3GD20H MEMS gyroscope model. Figure 2 shows its physical photo, and Table 1 summarizes the key parameters of this device.

This study adopted accelerated life testing under a vibration environment to accelerate the exposure of defects and weak points of test samples. The experimental setup consisted of a vibration table, a controller, and a data acquisition system. The vibration table provided a vibration load for the test sample [33]. The controller was used to set the experimental parameters, such as vibration frequency, amplitude and duration, etc. The performance data of the MEMS gyroscope during vibration were recorded by the data acquisition system. When there was a significant deviation in the output curve of the gyroscope, it was considered that the gyroscope was faulty. The main experimental steps are as follows:(1)First, we set the vibration stress level, such as frequency, amplitude, etc.(2)Subsequently, the L3GD20H MEMS gyroscope model was fixed on the vibration table, and the corresponding circuit and data acquisition equipment was connected to monitor and record the experimental data in real time.(3)Finally, we initiated the vibration equipment to apply vibrations of varying stress levels to the MEMS gyroscope according to preset parameters and recorded the data information.

Following the methodology outlined in this study, we investigated how the L3GD20H parameters degrade under vibrational stress levels of 30 g_n_, 40 g_n_, and 50 g_n_. Our primary goal was to measure the failure time of the MEMS gyroscope at each stress level. Table 2 presents the experimental results.

In addition, we analyzed the voltage output readings of the MEMS gyroscope rotor at different time intervals throughout the experiment. These results are summarized in Table 3.

Using the data from Table 2 and Table 3, we developed an accelerated degradation model for the MEMS gyroscope rotor’s zero bias scale parameter. Figure 3 shows the resulting accelerated degradation equation curve.

Figure 3 demonstrates a close match between the degradation curves obtained through our accelerated degradation modeling and the curves derived from the physical MEMS gyroscope rotor. A minor deviation within the 34–44 g_n_ stress range is observed, but this difference does not significantly affect the overall degradation pattern. This similarity validates the effectiveness of our proposed methodology in accurately simulating the zero bias scale parameter degradation of MEMS gyroscope rotors under vibrational conditions [34]. This accurate simulation is essential for the subsequent analysis of MEMS gyroscope degradation parameters.

We then used the distributional assumption method to illustrate the probability distribution of MEMS gyroscope rotor failure times under different vibrational stress levels, as shown in Figure 4.

Figure 4 reveals a correlation among stress level, failure time, and failure percentage of the MEMS gyroscope rotor. As stress levels increase, a reduction in failure time and an increase in failure percentage are observed. This indicates that a higher stress level accelerates the degradation of the rotor’s scale factor and bias parameters, thereby increasing the likelihood of failure. Accelerated life testing efficiently unveils device failures. Therefore, to enhance MEMS gyroscope reliability and stability, it is crucial to minimize stress levels and uphold appropriate usage and maintenance protocols [35,36].

Table 4 presents the estimated scale factor and zero bias parameters of the MEMS gyroscope rotor under different vibrational stress levels, where the scale factors are used as measurement indicators.

Table 4 reveals that, at the same stress level, the parameter *m* values for both the scale factor and zero bias are relatively small, while the parameter *η* values are larger. As stress levels increase, both the scale factor and zero bias parameters of the MEMS gyroscope rotor decrease, with *η* showing a more significant decline. This demonstrates the effectiveness of our proposed method in estimating the MEMS gyroscope rotor’s scale factor and zero bias parameters. These parameters provide key information for assessing the degradation of these critical performance metrics under vibrational conditions.

To further analyze the degradation behavior, we examine the degradation trajectory of the MEMS gyroscope rotor’s zero bias parameters under different vibrational stress levels, using the degradation percentage of the zero bias parameter for our metrics. The results are illustrated in Figure 5.

Figure 5 indicates that the degradation of the MEMS gyroscope rotor’s zero bias parameter worsens with increasing stress levels. During the initial 20 h, there is minimal difference in zero bias behavior across different stress levels. However, as time progresses, a clear correlation between stress level and the percentage change in the zero bias parameter is observed. This indicates that higher vibrational stress causes more significant zero bias changes within the same operational time, potentially shortening the MEMS gyroscope rotor’s lifespan.

Building upon the degradation data for the zero bias and scale factor parameters, we performed a reliability analysis of the MEMS gyroscope rotor. The results are presented in Figure 6.

As shown in Figure 6, the reliability of the MEMS gyroscope rotor decreases over time. Prior to 1.5 × 10^3^ h, the zero bias and scale factor reliability curves were in a stationary state. After 1.5 × 10^3^ h, the zero bias undergoes a notable decline, followed by a significant decrease in scale factor reliability after 1.8 × 10^3^ h. At approximately 2.25 × 10^3^ h, the degradation trend of zero bias and scale factor reliability slightly slows down. Both the zero bias and scale factor reliability reach 0.6 at the 4.0 × 10^3^ and 4.5 × 10^3^ h marks, respectively. Compared with the single parameter evaluation method proposed by Qin et al., the multi-parameter reliability evaluation method proposed in this paper accurately predicts the degradation trend of gyroscope rotor parameters [8]. Compared with the evaluation method introduced by He et al., the reliability evaluation method proposed in this paper performs better in predicting the trend of device reliability before 0.6 and more finely monitors the operating status of MEMS devices [9,37].

The experiment demonstrates that, as stress levels increase, the failure time of the MEMS gyroscope rotor diminishes gradually, accompanied by a rise in failure percentage. It is observed that the bias and scale factor parameters of MEMS gyroscopes gradually degrade under stress in vibrational environments, resulting in a declining trend in device reliability over time. A reliability assessment model for MEMS gyroscope rotor parameter degradation is established by employing distribution assumption methods and Copula functions. This model better predicts the evolving trend of device parameters over time, offering valuable insights for device maintenance and management. 

## 4. Conclusions

In the present work, a method for evaluating the reliability of MEMS gyroscope rotor parameter degradation under vibrational conditions was developed. We analyzed how vibrations affect these parameters and developed an accelerated degradation model using the distributional assumption method. To assess reliability, we integrated the Copula function in the proposed method. Experimental results confirmed the effectiveness of our approach in accurately capturing the changes in scale factor and zero bias reliability of the MEMS gyroscope rotor over time. During the actual operation of the MEMS gyroscope, the experimental data and the proposed evaluation method can be combined to continuously monitor the changes in the equipment parameters. By regularly collecting and analyzing data, the trend of MEMS gyroscope parameter degradation can be detected and corrected in time to ensure the stability of the equipment.

Building upon this work, future research will broaden its scope by studying different types of MEMS gyroscope samples. Moreover, upcoming investigations will delve into the internal mechanisms within MEMS gyroscopes that contribute to performance degradation. Understanding these mechanisms will facilitate the development of more effective optimization strategies. Additionally, innovative protection and maintenance techniques will be explored to ensure optimal MEMS gyroscope performance during operation. These further investigations aim to advance MEMS gyroscopes and provide practical guidance for their utilization.

## Figures and Tables

**Figure 1 micromachines-15-00648-f001:**
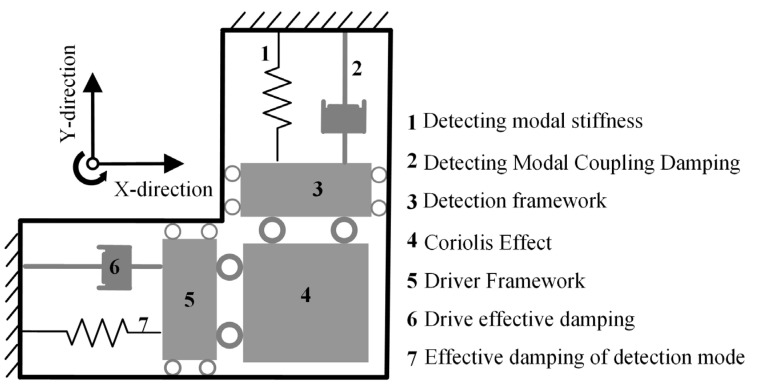
Schematic diagram of the MEMS gyroscope structure.

**Figure 2 micromachines-15-00648-f002:**
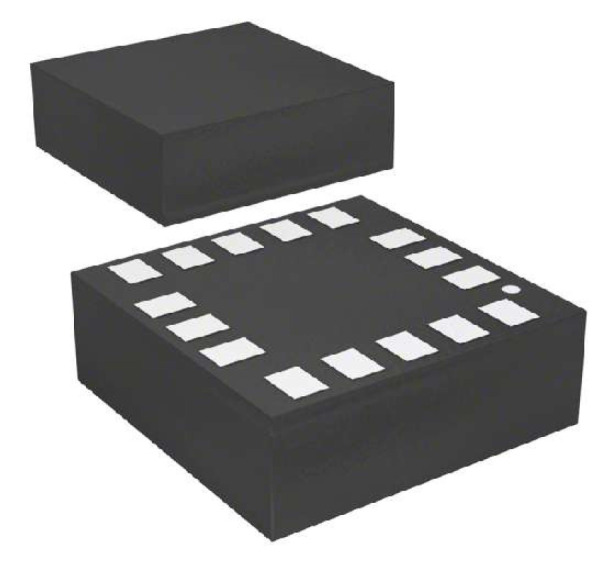
The L3GD20H MEMS gyroscope physical photo.

**Figure 3 micromachines-15-00648-f003:**
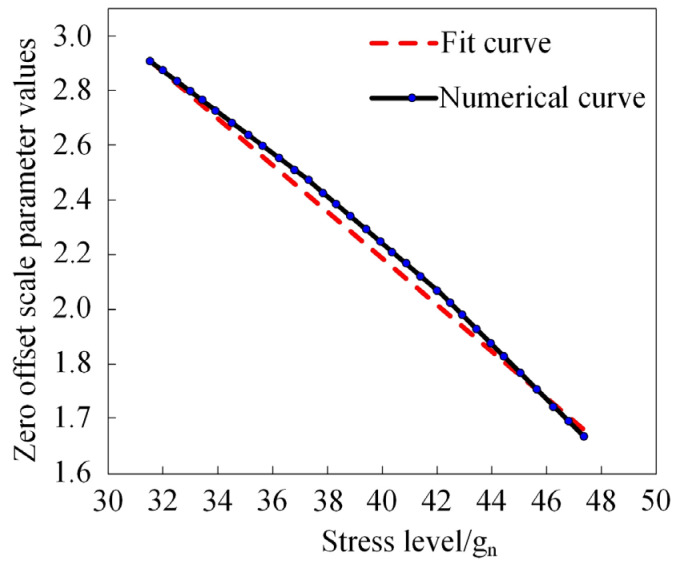
Degradation curve of zero offset scale parameters of MEMS gyroscope rotor.

**Figure 4 micromachines-15-00648-f004:**
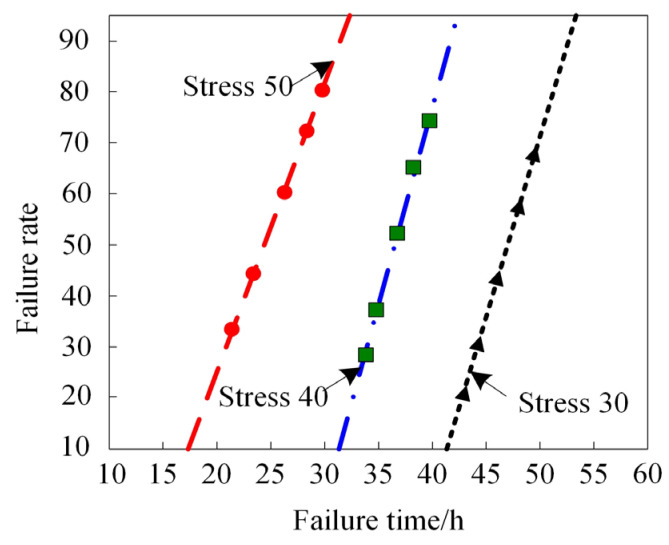
Failure time probability of MEMS gyroscope rotors at different stress levels.

**Figure 5 micromachines-15-00648-f005:**
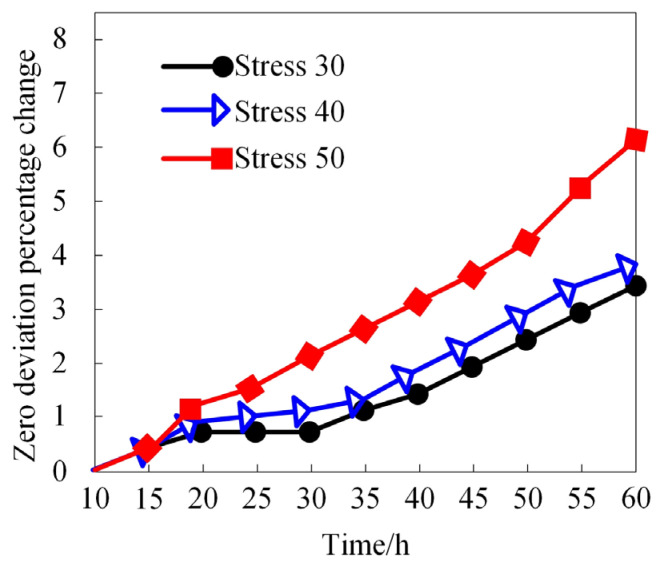
Estimated degradation trajectory of zero bias parameters under different vibrational stresses.

**Figure 6 micromachines-15-00648-f006:**
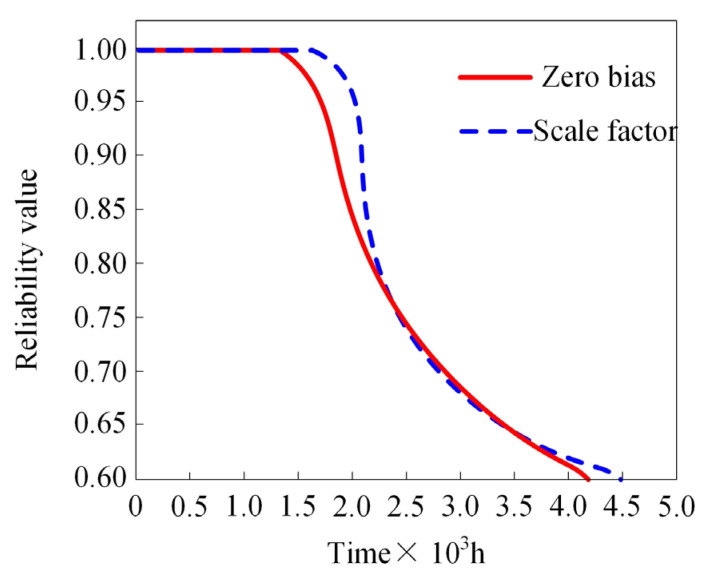
Reliability evaluation results of MEMS gyroscope rotor.

**Table 1 micromachines-15-00648-t001:** The L3GD20H MEMS gyroscope parameters.

Device Parameters	Numerical Value
package	LGA-16
resolution ratio	16bit
sensitivity	70 mdps/digit
Minimum operating temperature	−40 °C
Maximum operating temperature	85 °C
Minimum power supply voltage	2.2 V
Maximum power supply voltage	3.6 V
size	3 mm × 3 mm × 1 mm

**Table 2 micromachines-15-00648-t002:** MEMS gyroscope failure times under different stress levels.

Number of Tests	Stress Level		
	30	40	50
1	56	41	24
2	55	40	23
3	54	39	21
4	52	38	20
5	48	35	18
6	48	33	20
7	47	35	19
8	46	35	16
9	48	40	17
10	48	38	16
11	45	38	16
12	44	26	14

**Table 3 micromachines-15-00648-t003:** MEMS gyroscope rotor voltage output values at different times.

Number of Tests	Stress Level		
	30	40	50
1	2.411	2.518	2.597
2	2.432	2.506	2.600
3	2.404	2.524	2.596
4	2.408	2.578	2.535
5	2.412	2.541	2.563
6	2.420	2.650	2.565
7	2.413	2.529	2.576
8	2.427	2.584	2.577
9	2.433	2.546	2.549
10	2.438	2.554	2.547
11	2.409	2.583	2.588
12	2.411	2.623	2.549

**Table 4 micromachines-15-00648-t004:** Scale factor and zero bias parameter estimation results for MEMS gyroscope rotors.

Number of Tests		Stress Level		
		30	40	50
Scale factor	Parameter *m*	23.507	31.519	21.665
	Parameter *η*	47.219	36.884	18.579
Zero bias	Parameter *m*	24.911	28.535	21.439
	Parameter *η*	46.302	37.641	16.517

## Data Availability

Data is contained within the article.

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
