# Peer review of "Modeling and Reliability Analysis of MEMS Gyroscope Rotor Parameters under Vibrational Stress"

_micromachines, 2024, doi:10.3390/mi15050648_

Round 1

Reviewer 1 Report

Comments and Suggestions for Authors

In this manuscript, the authors develop an accelerated degradation model using the distributional assumption method, which focusing on zero bias and scale factor as key performance indicators. The method can effectively characterizes changes in the reliability of the MEMS gyroscope rotor's parameter degradation under vibration. The experiment shown that evaluation method is worthful in the technological application. however, there are some errors in the manuscript details. This manuscript can be accepted after addressing the following questions:

1. Section 2.2 seems incomplete and lacks sufficient details to fully understand the impact of vibration on the MEMS gyroscope rotor parameters. Please supplement relevant content.

2. Please provide more details on the experimental setup used to apply the vibrational stress levels to the MEMS gyroscope. This information is important for reproducibility.

3. In Table 1, more detailed features about the L3GD20H gyroscope should be provided, and the physical photos of the components should also be provided in the manuscript.

4. Line 105 page 3, it is mentioned that ' 16 rotors were selected and divided into four groups. These groups were induced to stress levels of 30gn, 40gn, and 50gn '. Please confirm whether the stress setting and grouping are correct.

5. Line 165 page 5, the formula H ( t ) should be written in italics, Please check the formatting of the formula throughout the manuscript.

6. Please provide higher resolution images for Figures 2-5 to improve readability, especially for the axis labels and legends.

Reviewer 2 Report

Comments and Suggestions for Authors

The subject of this manuscript (Modeling and Reliability Analysis of MEMS Gyroscope Rotor Parameters Under Vibrational Stress) is interesting and within the scope of Micromachines. A method for evaluating the reliability of parameter degradation under vibration to improve the effective use of MEMS gyroscopes. However, in its current state the manuscript contains several shortcomings (see below), more work is required at this stage.

1)The main problem with this manuscript is that parts of it read more like a technical report and lack the necessary quality for a research article.

For example, the introduction is very short and only five (5) references are cited in this section; a more detailed discussion of the results (including a comparison to available results from literature) is required!  Furthermore, the English should be carefully checked throughout the manuscript.

2) The conclusions are consistent with the presented data. However, a problem is, as also stated by the authors, that "...this study only uses the L3GD20H MEMS gyroscope model as the experimental object. The number of samples is small and cannot cover the characteristics of all samples, which may lead to deviations in the evaluation results..."

3) 5 figures and 4 tables of good quality are included. However, the tables should be renumbered, the second Table 2 should be Table 4.

4) EXPERIMENTAL ANALYSIS section:  Only two references [27, 28] are cited in this section.  A more detailed discussion of the results is required, comparing their results with in literature-reported data!

6) FIGURES & TABLES: 5 figures and 4 tables of good quality are included. However, the tables should be renumbered, the second Table 2 should be Table 4.

Comments on the Quality of English Language

Please carefully check the English throughout the manuscript. It does not always meet the journal’s desired standard. 

Round 2

Reviewer 2 Report

Comments and Suggestions for Authors

The authors have done a good job and have carried out all the suggested changes, i.e. the revised manuscript can be accepted in its current form.